

# Let's be pals again: major systematic changes in Palaemonidae (Crustacea: Decapoda)

Sammy De Grave[1], Charles H.J.M. Fransen[2] and Timothy J. Page[3,4]

[1] Oxford University Museum of Natural History, Oxford University, Oxford, United Kingdom
[2] Department of Marine Zoology, Naturalis Biodiversity Center, Leiden, The Netherlands
[3] Australian Rivers Institute, Griffith University, Nathan, QLD, Australia
[4] Water Planning Ecology, Queensland Department of Science, Information Technology and Innovation, Dutton Park, QLD, Australia

## ABSTRACT

In recent years the systematic position of genera in the shrimp families Gnathophyllidae and Hymenoceridae has been under debate, with phylogenetic studies suggesting the families are not real family level units. Here, we review the molecular evidence as well as the morphological characters used to distinguish both families, leading to the conclusion that neither family is valid. Further, we studied the structural details of the single morphological character which distinguishes the two subfamilies (Palaemoninae, Pontoniinae) in Palaemonidae, as well as their phylogenetic relationship. As the supposed character distinction plainly does not hold true and supported by the phylogenetic results, the recognition of subfamilies in Palaemonidae is not warranted. As a consequence, all three supra-generic taxa (Gnathophyllidae, Hymenoceridae, Pontoniinae) are thus herein formally synonymised with Palaemonidae.

Corresponding author
Sammy De Grave,
sammy.degrave@oum.ox.ac.uk

## INTRODUCTION

In recent years, the higher level of systematics of caridean shrimps has seen considerable changes at subfamily, family and superfamily level, but not without controversy. To take but one example, *Bracken, De Grave & Felder (2009)* suggested that the family Oplophoridae could be polyphyletic, however this study only included 4 genera (out of 10). This was followed by *Chan et al. (2010)* who, on the basis of a molecular phylogeny of 10 species from 9 genera, split the family into two families, Oplophoridae and Acanthephyridae, underpinned by habitat and morphological differences between the two families. However, *Wong et al. (2015)* in a more comprehensive study of 30 species in 9 genera, consider the family to be monophyletic, yet comprising two distinct clades, which correspond to the above separate families. Finally, *Aznar-Cormano et al. (2015)* in a wide-ranging analysis with coverage across all caridean families, recovered both families as distinct lineages with high support, but with poorly resolved relationships between them.

Despite such problems, currently 39 families of caridean shrimps are recognised (*De Grave & Fransen, 2011*; *Baeza et al., 2014*; *De Grave et al., 2014*). Seven of these families used

to be placed in the superfamily Palaemonoidea Rafinesque, 1815 (see *De Grave & Fransen, 2011*), namely Anchistioididae Borradaile, 1915; Desmocarididae Borradaile, 1915; Euryrhynchidae Holthuis, 1950; Gnathophyllidae Dana, 1852; Hymenoceridae Ortmann, 1890; Palaemonidae Rafinesque, 1815 and Typhlocarididae Annandale & Kemp, 1913. In previous classifications (e.g., *De Grave et al., 2009*; *De Grave & Fransen, 2011*) a further family was recognised, Kakaducarididae *Bruce, 1993*. Following the phylogenetic analysis in *Page et al. (2008)*, *Short, Humphrey & Page (2013)* in a morphological reappraisal relegated this family to the synonymy of Palaemonidae. Although Palaemonoidea at superfamily level appears to indeed form a monophyletic group (*Li et al., 2011*), superfamilies are not often formally used any more in caridean systematics, and we herein refer to this assemblage of families as the palaemonoid clade. Traditionally, Palaemonidae has been thought to comprise two subfamilies, Palaemoninae Rafinesque, 1815 (primarily freshwater and temperate coastal species) and Pontoniinae *Kingsley, 1879* (primarily tropical species, most abundant on coral reefs), although the morphological dividing line between both can be rather arbitrary (*Bruce, 1995*).

In common with several other taxa, the systematic composition of the palaemonoid clade has been somewhat mired in controversy in recent decades. It was not until *Chace (1992)* that Hymenoceridae were recognised as separate from Gnathophyllidae. In contrast, Typhlocarididae was comprised of two subfamilies in his classification, Typhlocaridinae and Euryrhynchinae, therein followed by the major compilations of *Chace & Bruce (1993)* and *Holthuis (1993)*. *Bruce (1993)* expressed the opinion that both these taxa are not closely related and should be treated as independent families, a view corroborated by the morphological discussion in *De Grave (2007)*.

*Mitsuhashi et al. (2007)* were the first to demonstrate that Gnathophyllidae, Hymenoceridae and Pontoniinae form a paraphyletic clade in their 18S/28S analysis of a limited dataset (only including 17 species from four families) and pointed out the congruence of larval morphology to this result. *Kou et al. (2013a)* expanded on this dataset (16S/18S/28S), with 44 species (7 families), but with heavy bias towards Palaemoninae (only 2 Pontoniinae were included). Despite this unbalanced sampling scheme, their results demonstrate Palaemoninae to be polyphyletic and the same paraphyletic assemblage of Gnathophyllidae, Hymenoceridae and Pontoniinae. Recently, *Gan et al. (2015)* provided yet one more variant, based on a combined analysis of 16S/H3/Nak/Enolase, with a heavy inclusion of Pontoniinae over Palaemoninae (as well as Gnathophyllidae, Hymenoceridae, Anchistioididae), but exclusive of the Atlantic families, Desmocarididae, Euryrhynchidae and Typhlocarididae. Nevertheless, their analysis once again recovers Gnathophyllidae and Hymenoceridae inside Pontoniinae. Despite this wealth of data, analyses to date have not included the full breadth of available molecular diversity within the palaemonoid clade as a whole, thus any systematic conclusions are at best partial, and at worst misleading. This has, in part, been due to the fact that different loci have often been sequenced for the different taxa, making a comparison between them impossible. We have trawled through available molecular data to assemble datasets that represent the lion's share of the currently available molecular

diversity within each of the nine suprageneric palaemonoid taxa (7 families, 2 subfamilies) so as to assess the relationships amongst them with fullest possible data.

The systematic distinction of the two subfamilies within Palaemonidae, i.e., Palaemoninae and Pontoniinae, has received scant scrutiny and has been generally followed without query. To date, no phylogenetic study has included sufficient taxa from both to allow a discussion of the validity of either subfamily. Morphologically, they are distinguished on a single character, of somewhat dubious validity. *Kingsley (1879)* distinguished both taxa on the basis of the presence/absence of a mandibular palp (therein followed by *Spence Bate, 1888*, a clearly variable character within each subfamily (see *Chace & Bruce, 1993*; *De Grave & Ashelby, 2013*). *Sollaud (1910)* distinguished both taxa on the basis of the presence/absence of a pleurobranch on the third thoracic somite, to which *Balss (1957)* added the ornamentation of the posterior telson. *Abele & Felgenhauer (1986)*, as well as *Bruce (1995)* showed the absence of a pleurobranch on the third maxilliped in selected species of *Macrobrachium*. Further, *Bruce (2002)* demonstrated variability in the size of the arthrobranch, which is reduced in *Leander manningi*. We therefore concur with *Bruce (1995)* that it is likely that both Palaemoninae and Pontoniinae have five pairs of pleurobranchs, leaving only the telson distinction. *Holthuis (1993)* defined the latter as follows: telson with two pairs of posterior "spines" and with one or more pairs of hairs (i.e., plumose setae)—Palaemoninae, versus telson usually with three pairs of posterior "spines"—Pontoniinae. However, *Bruce (1995)* drew attention to the fact that in many Pontoniinae, the submedian "spines" are often also plumose. In the present contribution, we provide a detailed morphological examination of these setae, in combination with molecular analyses to investigate the relationships of the two subfamilies.

## MATERIAL AND METHODS

### Dataset construction for molecular analysis

Genbank (www.ncbi.nlm.nih.gov) was searched for sequences of palaemonoid taxa on 24 November, 2014. We were looking for genetic markers for which there were data from all seven palaemonoid families (Anchistioididae, Desmocarididae, Euryrhynchidae, Gnathophyllidae, Hymenoceridae, Palaemonidae, Typhlocarididae) and for which there was also good coverage of genera of the two subfamilies within Palaemonidae (Palaemoninae, Pontoniinae). In particular, we strove to include the various clades and divergent taxa within each subfamily as identified in previous restricted subfamily studies (*Ashelby et al., 2012*; *Kou et al., 2013b*; *Gan et al., 2015*). We only included species for which there were at least two different independent markers. It quickly became apparent that some loci were available only for one subfamily (e.g., Pontoniinae—Enolase, NaK, Pepck), and so were not informative across all taxa. The four markers that had the best coverage across all taxa were the mitochondrial 5' cytochrome *c* oxidase I (COI), mitochondrial 16S ribosomal DNA (16S), nuclear Histone 3 (H3) and nuclear 18S ribosomal DNA (18S). Preliminary analyses of COI data quickly established that although it was effective at grouping very closely related species, it was highly ineffective at inferring deeper systematic relationships, which is unsurprising given its relatively rapid rate of molecular divergence.

Thus, we settled on 16S, H3 and 18S for our analyses, as this combination of markers with differing levels of divergence may pull out any strong systematic relationships.

Relevant data from GenBank, and an additional three new H3 sequences of our own (*Gnathophylloides mineri, Manipontonia psamathe, Pontonia manningi*) were combined (Table 1), with the alpheid *Betaeus longidactylus* as an outgroup. Sequences of the three markers were imported into Mega 6 (*Tamura et al., 2013*) and each aligned separately using Muscle (*Edgar, 2004*) within Mega. The most appropriate substitution model (lowest Bayesian Information Criterion score) was chosen with Mega. Four separate datasets were created; 16S (424 base pairs [bp], 45 species); H3 (327 bp, 42 species); 18S (1559 bp, 23 species), combined 16S/H3/18S (2310 bp, 45 species), with any unavailable data coded as missing (Table 1).

## Molecular analyses

The single marker datasets were analysed using Bayesian analyses in MrBayes 3.2 (*Ronquist et al., 2012*) and Maximum Likelihood in Mega (bootstrapped 1000 times), both using the relevant molecular model for each marker. The Bayesian analyses were done using the following parameters: 5 million generations, trees sampled every 1000 cycles, 25% burn in, two runs of four chains heated to 0.2. The combined dataset was analysed using Bayesian analyses as above.

Formal phylogenetic support for various systematic schemes was assessed by constraining the topology of the Bayesian analyses in the relevant way and then rerunning MrBayes for each dataset. Constrained versus unconstrained harmonic means of log likelihood values were then compared with Bayes Factors (*Kass & Raftery, 1995*). Seven different topological constraints were tested (the last 5 only on the combined dataset), with no constraints place on topologies within each defined clade unless specified:

(A) species of Palaemoninae form a clade, and species of Pontoniinae form a separate clade; (B) species of Palaemoninae form one clade, and species of Gnathophyllidae/Hymenoceridae/Pontoniinae form a separate single clade; (C) species of Palaemonidae/Gnathophyllidae/Hymenoceridae form a clade; (D) Palaemonidae form a clade, and within it both Palaemoninae and Pontoniinae are reciprocally monophyletic (effectively the current state of play); (E) Palaemonidae form a clade, and within it Palaemoninae forms a clade sister to a clade of Pontoniinae/Gnathophyllidae/Hymenoceridae; (F) species of Anchistioididae/Palaemonidae/Gnathophyllidae/Hymenoceridae form one clade; (G) species of Desmocarididae/Euryrhynchidae/Palaemonidae/Gnathophyllidae/Hymenoceridae form a clade.

## Morphological study

Twelve species (Table 2) were selected randomly from Palaemoninae (4 species) and Pontoniinae (8 species) to investigate the posterior ornamentation of the telson, which currently is the only morphological character which distinguishes both subfamilies. Tissue preparation for Scanning Electron Microscopy (SEM) follows *De Grave & Wood (2011)*, whereby tissue is hydrated to distilled water via a series of graded ethanol solutions, briefly sonicated using a light surfactant and dehydrated in graded ethanol to 100%. Drying was

**Table 1** Details of sequences used in this study.

| | Genbank accession numbers | | |
| --- | --- | --- | --- |
| | **16S** | **H3** | **18S** |
| **Anchistioididae** | | | |
| *Anchistioides antiguensis* (Schmitt, 1924) | EU920911 | EU921043 | EU920936 |
| *Anchistioides* willeyi (Borradaile, 1900) | KC515030 | KC515074 | – |
| **Desmocarididae** | | | |
| *Desmocaris* sp. | EU868651 | – | EU868742 |
| **Euryrhynchidae** | | | |
| *Euryrhynchus wrzesniowskii* Miers, 1877 | EU868654 | – | EU868745 |
| **Gnathophyllidae** | | | |
| *Gnathophylloides mineri* Schmitt, 1933 | EU868659 | KT224392 | EU868750 |
| *Gnathophyllum americanum* Guérin–Meneville, 1855 | EU868660 | JF346317 | EU868751 |
| **Hymenoceridae** | | | |
| *Hymenocera picta* Dana, 1852 | EU868663 | JF346328 | EU868754 |
| *Phyllognathia ceratophthalma* (Balss, 1913) | KC515032 | KC515076 | DQ642847 |
| **Palaemonidae–Palaemoninae** | | | |
| *Arachnochium mirabilis* (Kemp, 1917) | KC515033 | KC515077 | KC515052 |
| *Brachycarpus biunguiculatus* (Lucas, 1846) | EU868684 | JN674391 | EU868779 |
| *Creaseria morleyi* (Creaser, 1936) | EU868688 | DQ079671 | DQ079746 |
| *Cryphiops caementarius* (Molina, 1782) | DQ079711 | DQ079672 | DQ079747 |
| *Leander tenuicornis* (Say, 1818) | EU868690 | JN674388 | EU868783 |
| *Leandrites deschampsi* (Nobili, 1903) | KC515039 | KC515081 | – |
| *Leptocarpus potamiscus* (Kemp, 1917) | JN674328 | JN674392 | – |
| *Macrobrachium rosenbergii* (De Man, 1879) | FM986637 | FM958123 | DQ642856 |
| *Nematopalaemon tenuipes* (Henderson, 1893) | KC515042 | JN674382 | – |
| *Palaemon concinnus* Dana, 1852 | KC515043 | KC515085 | KC515056 |
| *Palaemon elegans* Rathke, 1837 | EU868696 | DQ079696 | DQ079764 |
| *Palaemon pandaliformis* (Stimpson, 1871) | JN674341 | JN674364 | – |
| *Urocaridella pulchella* Yokes & Galil, 2006 | KC515050 | KC515092 | KC515062 |
| **Palaemonidae–Pontoniinae** | | | |
| *Anchiopontonia hurii* (Holthuis, 1961) | KF738358 | KF738309 | – |
| *Anchistus custos* (Forskål, 1775) | KF738360 | KF738311 | – |
| *Conchodytes meleagrinae* Peters, 1852 | KC515051 | KC515093 | EF540837 |
| *Coralliocaris graminea* (Dana, 1852) | KF738361 | KF738313 | AM083319 |
| *Cuapetes andamanensis* (Kemp, 1922) | JX025214 | KF738315 | – |
| *Cuapetes elegans* (Paulson, 1875) | JX025213 | KF738316 | – |
| *Dactylonia ascidicola* (Borradaile, 1898) | KF738363 | KF738317 | – |
| *Harpiliopsis spinigera* (Ortmann, 1890) | JX025206 | KF738319 | – |
| *Harpilius lutescens* Dana, 1852 | JX025205 | KF738320 | – |
| *Ischnopontonia lophos* (Barnard, 1962) | KF738364 | KF738321 | |
| *Laomenes nudirostris* (Bruce, 1968) | KF738366 | KF738323 | – |
| *Manipontonia psamathe* (De Man, 1902) | JX025199 | KT224393 | – |
| *Palaemonella spinulata* Yokoya, 1936 | KF738367 | KF738325 | – |

Table 1 (*continued*)

| | Genbank accession numbers | | |
|---|---|---|---|
| | **16S** | **H3** | **18S** |
| *Periclimenaeus bidentatus* Bruce, 1970 | KF738368 | KF738326 | – |
| *Periclimenes brevicarpalis* (Schenkel, 1902) | JX025191 | JF346324 | JF346254 |
| *Periclimenes calcaratus* Chace & Bruce, 1993 | KF738370 | KF738329 | – |
| *Philarius gerlachei* (Nobili, 1905) | JX025177 | KF738333 | – |
| *Platycaris latirostris* Holthuis, 1952 | KF738371 | KF738335 | – |
| *Pliopontonia furtiva* Bruce, 1973 | KF738372 | KF738336 | – |
| *Pontonia manningi* Fransen, 2000 | EU868705 | KT224394 | EU868800 |
| *Thaumastocaris streptopus* Kemp, 1922 | KF738373 | KF738337 | DQ642852 |
| *Zenopontonia soror* (Nobili, 1904) | JX025178 | KF738332 | – |
| **Typhlocarididae** | | | |
| *Typhlocaris salentina* Caroli, 1923 | EU868713 | – | EU868808 |
| **Alpheidae** | | | |
| *Betaeus longidactylus* Lockington, 1877 | JX010752 | JX010771 | JF346263 |

**Table 2** Species examined by SEM for morphology of telson setation (all material is accessioned in the Oxford University Museum of Natural History-OUMNH.ZC).

| | Origin | Accession number |
|---|---|---|
| **Palaemonidae–Palaemoninae** | | |
| *Leander tenuicornis* (Say, 1818) | USA | OUMNH.ZC.2006-11-007 |
| *Macrobrachium amazonicum* (Heller, 1862) | Brazil | OUMNH.ZC.2002-27-003 |
| *Palaemon adspersus* Rathke, 1837 | Greece | OUMNH.ZC.2003-03-001 |
| *Palaemon modestus* (Heller, 1862) | Kazahkstan | OUMNH.ZC.2012-01-068 |
| **Palaemonidae–Pontoniinae** | | |
| *Conchodytes nipponensis* (De Haan, 1844) | Japan | OUMNH.ZC.2011-11-001 |
| *Cuapetes americanus* (Kingsley, 1878) | Panama | OUMNH.ZC.2003-33-050 |
| *Jocaste lucina* (Nobili, 1901) | Chagos | OUMNH.ZC.2014-09-038 |
| *Palaemonella rotumana* (Borradaile, 1898) | Singapore | OUMNH.ZC.2011-02-003 |
| *Periclimenaeus caraibicus* Holthuis, 1951 | Panama | OUMNH.ZC.2008-14-065 |
| *Periclimenes brevicarpalis* (Schenkel, 1902) | Taiwan | OUMNH.ZC.2010-02-003 |
| *Stegopontonia commensalis* Nobili, 1906 | Taiwan | OUMNH.ZC.2010-02-039 |
| *Thaumastocaris streptopus* Kemp, 1922 | Israel | OUMNH.ZC.2011-05-024 |

achieved using the HMDS method, and specimens coated with a gold-palladium mixture using an E5000 sputter coater. Mounted specimens were observed and photographed using a JEOL JSM-5510 microscope; images were not post processed with image software. SEM observations were complemented by light microscopy of a much wider range of species to verify the results. Setal terminology in general follows *Garm (2004)*, although we consider the term cuspidate to also include more elongated forms of setae termed "intermediate form between cuspidate and simple" in *Garm (2004)* to facilitate discussion.

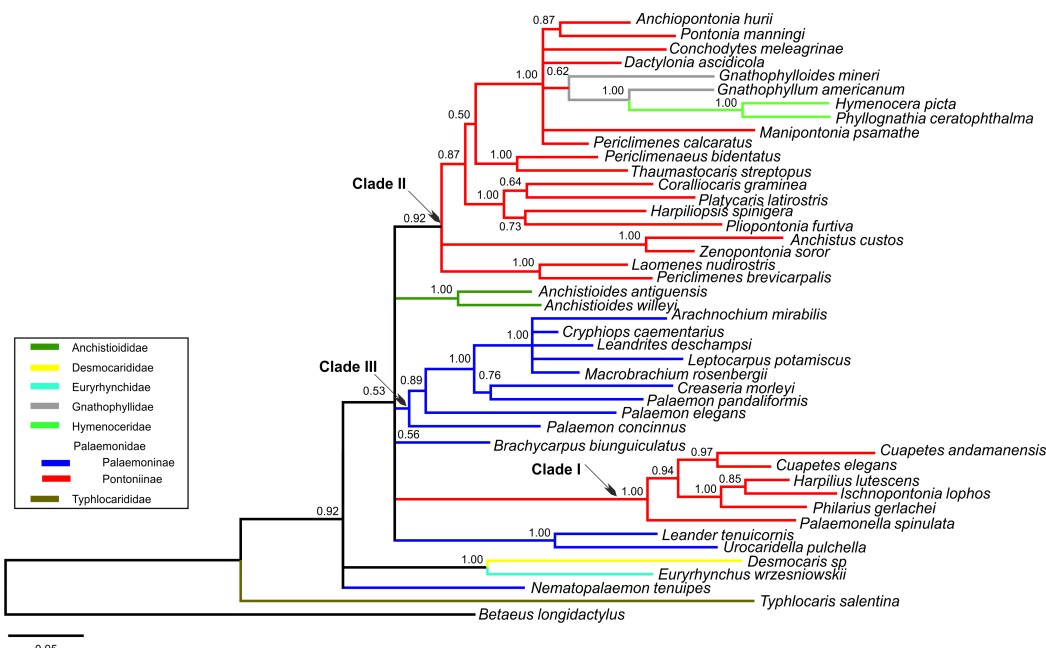

**Figure 1 Bayesian majority rule consensus topology for combined dataset (16S/H3/18S) of the palaemonoid clade.** No constraints, only clades with >0.50 posterior probability are shown, Tree Score = −16540.11. For definitions of palaemonid clades, see text.

# RESULTS

## Molecular results

Bayesian (BA) trees were produced for the combined dataset (Fig. 1, Figs. S1 and S2), as well as Bayesian and Maximum Likelihood (ML) trees for each single locus dataset (Figs. S3–S8). Majority rule consensus trees are displayed for Bayesian trees (i.e., clades >0.50 posterior probability). BA and ML analyses inferred similar clades at shallower levels for the single marker datasets, with ML support values generally lower. Our analyses are primarily based on the combined dataset, with the single locus analyses provided for reference.

### Typhlocarididae

Typhlocarididae was represented by two of the three markers (16S, 18S). Its sole genus was recovered strongly as sister to all other palaemonoids (Fig. 1). Both its 16S and 18S sequences were highly differentiated from the rest of the sampled Palaemonoidea.

### Anchistioididae

The two species from the sole anchistioidid genus (*Anchistioides*) formed a strong clade in all analyses. Its relationship with the other families (excluding Typhlocarididae) is not immediately apparent. It did not form strong clades with the other families in most analyses, except in both 18S analyses where it groups with Palaemonidae, Gnathophyllidae and Hymenoceridae to the exclusion of Desmocarididae and Euryrhynchidae. However when Anchistioididae was constrained to form a clade with Palaemonidae/Gnathophyllidae/Hymenoceridae in a combined analysis (Constraint F), its

**Table 3** Datasets, molecular models and tree scores for analyses conducted in this study.

| Dataset | Molecular model | N | Figure | Tree Scores | Difference versus unconstraint | BF strength of evidence of difference |
|---|---|---|---|---|---|---|
| **16S/H3/18S (Combined)** | | 45 | | | | |
| Unconstrained BA | | | 1 | −16540.11 | | |
| Constraint A | | | | −16617.59 | 77.48 | Very strong against |
| Constraint B | | | S1 | −16546.48 | 6.37 | Very strong against |
| Constraint C | | | S2 | −16540.26 | 0.15 | Equivocal |
| Constraint D | | | | −16616.49 | 76.38 | Very strong against |
| Constraint E | | | | −16544.29 | 4.18 | Strong |
| Constraint F | | | | −16547.56 | 7.45 | Very strong against |
| Constraint G | | | | −16545.17 | 5.06 | Very strong against |
| **16S rDNA (16S)** | TN93+G+I | 45 | | | | |
| Unconstrained ML | | | S3 | −7434.59 | | |
| Unconstrained BA | | | S4 | −8098.11 | | |
| Constraint A | | | | −8133.18 | 35.07 | Very strong against |
| Constraint B | | | | −8120.15 | 22.04 | Very strong against |
| **Histone 3 (H3)** | K2+G+I | 42 | | | | |
| Unconstrained ML | | | S5 | −3329.51 | | |
| Unconstrained BA | | | S6 | −3364.55 | | |
| Constraint A | | | | −3386.06 | 21.51 | Very strong against |
| Constraint B | | | | −3371.81 | 7.26 | Very strong against |
| **18S rDNA (18S)** | K2+G+I | 23 | | | | |
| Unconstrained ML | | | S7 | −4812.79 | | |
| Unconstrained BA | | | S8 | −4859.85 | | |
| Constraint A | | | | −4907.84 | 47.99 | Very strong against |
| Constraint B | | | | −4872.23 | 12.38 | Very strong against |

**Notes.**

BA, Bayesian analysis; BF, Bayes Factor; G, Gamma Rate Distribution; I, Invariant sites; K2, Kimua 2-parameter; ML, Maximum Likelihood; TN93, Tamara-Nei model.

score was slightly worse than when Desmocarididae and Euryrhynchidae were constrained to form a clade with Palaemonidae/Gnathophyllidae/Hymenoceridae (Constraint G) (Table 3), so its precise relationship with the other families is unclear.

### *Desmocarididae and Euryrhynchidae*

Desmocarididae and Euryrhynchidae were represented by two of the three markers (16S, 18S). They formed a strong clade with each other (Fig. 1), in particular due to their 18S data, but their relationship to the other families (except Typhlocarididae) is unclear in the same way as Anchistioididae above. They may be sister to Anchistioididae/Palaemonidae/Gnathophyllidae/Hymenoceridae (Fig. S2), however the tree score when they are forced to form a clade with Palaemonidae/Gnathophyllidae/Hymenoceridae to the exclusion of Anchistioididae (Constraint G) is marginally better than when Anchistioididae is constrained to Palaemonidae/Gnathophyllidae/Hymenoceridae (Constraint F) (Table 3), however the difference is not great, and neither constraint produces a particularly bad score relative to the unconstrained analysis. Therefore the relationship of the clade formed by Desmocarididae/Euryrhynchidae is unclear

relative to Anchistioididae/Palaemonidae/Gnathophyllidae/Hymenoceridae. However, when constraints are applied to other taxa (Constraints B, C), Anchistioididae forms a strong clade with palaemonid taxa to the exclusion of Desmocarididae/Euryrhynchidae (Figs. S1 and S2), so it is possible that Desmocarididae/Euryrhynchidae is sister to Anchistioididae/Palaemonidae/Gnathophyllidae/Hymenoceridae, but more data is required to explore this further.

### Palaemonidae

As currently defined a pure Palaemonidae is not supported as a distinct separate unit (Constraint D) since Gnathophyllidae and Hymenoceridae clearly nest within, making Palaemonidae paraphyletic at best (Fig. 1). However when one includes Gnathophyllidae and Hymenoceridae within Palaemonidae and does not enforce monophyly of the subfamilies, then there is little difference compared to completely unconstrained analyses (Constraint C). Even when Palaemoninae and Pontoniinae/Gnathophyllidae/Hymenoceridae are constrained to be sisters within a monophyletic Palaemonidae, the resulting tree score is not really too much worse (Constraint E, Table 3). This implies that the "problem" is with the internal structure of Palaemonidae rather than in its relationship to others, and that it may well be a monophyletic unit. However as stated above, it is also unclear how Desmocarididae/Euryrhynchidae, and particularly Anchistioididae, relate to Palaemonidae. Plainly Palaemonidae contains Gnathophyllidae and Hymenoceridae. However, it is not yet clear phylogenetically whether Palaemonidae is truly monophyletic relative to Anchistioididae and/or Desmocarididae/Euryrhynchidae, within Palaemonoidea.

### Palaemoninae and Pontoniinae

I Palaemoninae and Pontoniinae do not form clear monophyletic clades in any of our analyses. When they are each constrained to monophyly (Constraint A), the scores are all very much worse than when unconstrained (Table 3). When Gnathophyllidae and Hymenoceridae are considered honorary Pontoniinae (Constraint B), then the tree scores improved markedly in all analyses (Table 3) (Fig. S1), but the evidence against this topology is still very strong. When Palaemoninae and Pontoniinae are constrained to clades within a monophyletic Palaemonidae (Constraint D), which is essentially the current taxonomy, the scores are very poor and so are unlikely to reflect phylogenetic reality. However when Gnathophyllidae and Hymenoceridae are included within Pontoniinae within a monophyletic Palaemonidae (Constraint E), then scores improve greatly, although still worse than unconstrained. The one constraint that approaches the unconstrained scores is when all species of Palaemoninae, Pontoniinae, Gnathophyllidae and Hymenoceridae are thrown together into a single clade but without any intraclade constraints (Constraint C) (Fig. S2).

Instead of a clear delineation of reciprocally monophyletic Palaemoninae and Pontoniinae (which includes Gnathophyllidae/Hymenoceridae), what emerges are a number of larger clades that contain either species of Palaemoninae or species of Pontoniinae/Gnathophyllidae/Hymenoceridae, and a few divergent species of Palaemoninae whose relationship is unclear (Fig. 1). However these clades and species do not form larger clades that equate to the subfamilies as currently defined.

### Morphology of the posterior margin of the telson in Palaemonidae

The posterior margin of the telson in the majority of Palaemoninae comprises a lateral pair of short cuspidate setae, a submedian pair of elongated, cuspidate setae and one or more pairs of median plumose setae (Figs. 2A, 2C and 2E). The plumose setae are classical in structure, with two rows of long setules, weakly articulated with the setal shaft (Figs. 2B and 2D). Although the examples shown herein (*Palaemon adspersus*, *Macrobrachium amazonicum*, *Leander tenuicornis*) only have a single pair of median plumose setae, several other taxa harbour two (e.g., *Brachycarpus biunguiculatus*) or more pairs (e.g., *Neopalaemon nahuatlus*, *Palaemon tonkinensis*). As exemplified herein by *Palaemon modestus* (Fig. 2F), deviations of this bauplan exist, with the species previously assigned to *Exopalaemon* (recently transferred to *Palaemon*), having lost the median plumose setae.

Although the extensive bauplan modification in Pontoniinae due to their extensive commensal relationships has resulted in more variation in the ornamentation of the posterior margin of the telson, many genera remain morphologically very similar in this respect to Palaemoninae. For example, the free living *Palaemonella rotumana* (Figs. 3A and 3B) and *Cuapetes americanus* (Figs. 3C and 3D) have a similar arrangement with a pair of lateral cuspidate setae, a submedian pair of elongated, cuspidate setae and a median pair of plumose setae. The median plumose setae are however more robust than their counterparts in Palaemoninae, with sparser and somewhat shorter setules. In some commensal taxa, the submedian pair of cuspidate setae is considerably shorter, as exemplified herein by the anemone associate, *Periclimenes brevicarpalis* (Fig. 3E) and the sponge associate *Periclimenaeus caraibicus* (Fig. 3F). Both, however, harbour a robust pair of median plumose setae, with somewhat shorter and sparser setules. In contrast, rather densely plumose median setae are evident in the coral dwelling *Jocaste lucina* (Figs. 4A and 4B) and the sponge associated *Thaumastocaris streptopus* (Figs. 4C and 4D), with the setules in both being as long as in Palaemoninae. In the morphologically highly modified, echinoid associated *Stegopontonia commensalis*, the median setae are very robust, but continue to display a reduced set of setules, both sparse (mainly restricted to basal part) and very short (Fig. 4E). A barely discernible set of minute setules is still present on the median setae in the equally highly modified, bivalve associate, *Conchodytes nipponensis* (Fig. 4F), which otherwise has only two pairs of extremely short and robust cuspidate setae, homologous to the median and submedian pairs in the other species.

## DISCUSSION

The numerous molecular analyses presented herein agree strongly in some respects, but agree only weakly in some, and even disagree in others. Therefore it is not always possible to come to unequivocal conclusions in all cases, but our overall hypothesis of relationships, based on the current molecular analyses in the palaemonoid clade is presented in Fig. 5. Available data suggests strongly that Typhlocarididae are a sister group to the rest of Palaemonoidea. Next, there is weak evidence that a clade of Desmocarididae/Euryrhynchidae are a sister group to the remaining taxa, however this is not certain. Anchistioididae may form a clade with Palaemonidae/Gnathophyllidae/Hymenoceridae,

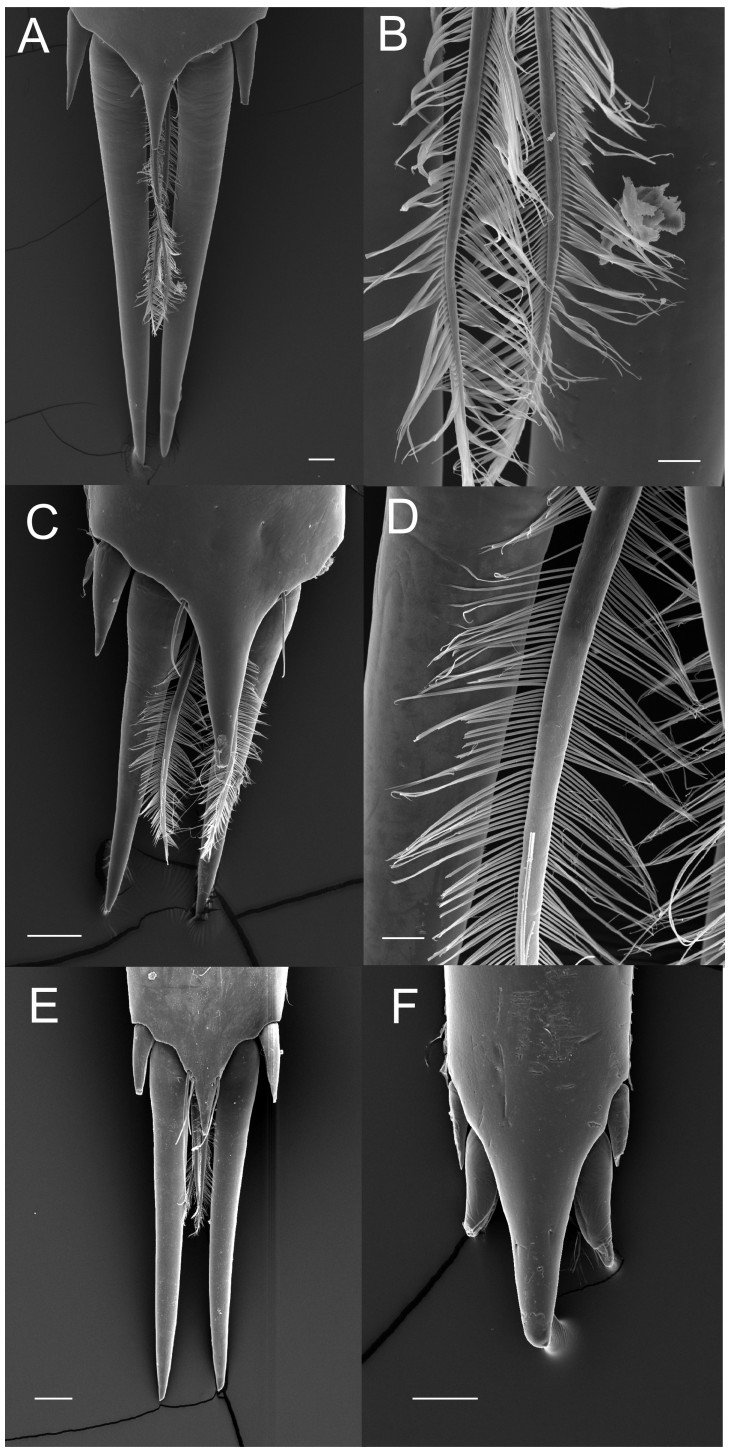

**Figure 2 Ornamentation of the posterior telson margin of some Palaemoninae.** (A) *Palaemon adspersus*; (B) same, detail of median setae; (C) *Macrobrachium amazonicum*; (D) same, detail of median setae; (E) *Leander tenuicornis*; (F) *Palaemon modestus*. Scale bars indicate 100 μm (A, C, D–E), 40 μm (B) or 20 μm (D).

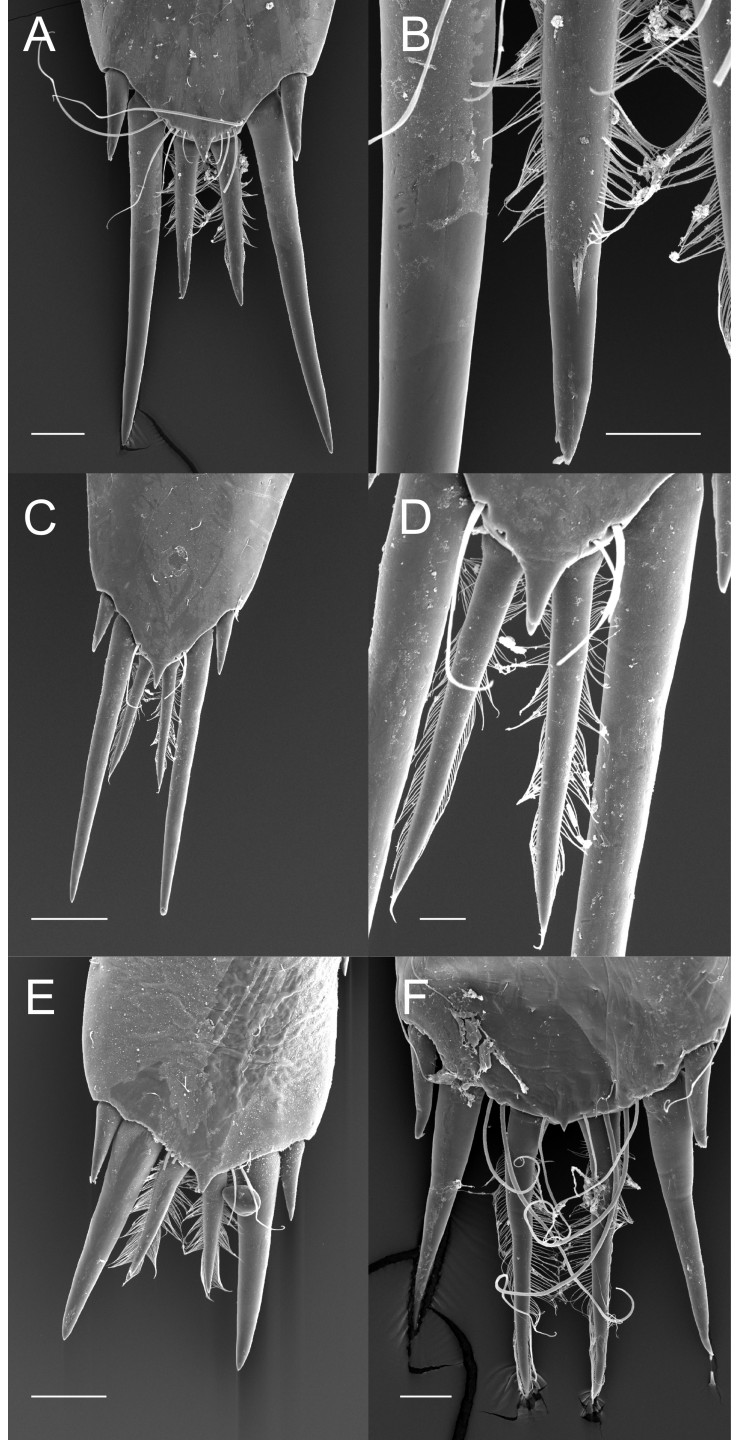

**Figure 3 Ornamentation of the posterior telson margin of some Pontoniinae.** (A) *Palaemonella rotumana*; (B) same, detail of median setae; (C) *Cuapetes americanus*; (D) detail of median setae; (E) *Periclimenes brevicarpalis*; (F) *Periclimenaeus caraibicus*. Scale bars indicate 100 μm (A, C, E), 50 μm (B, F) or 20 μm (D).

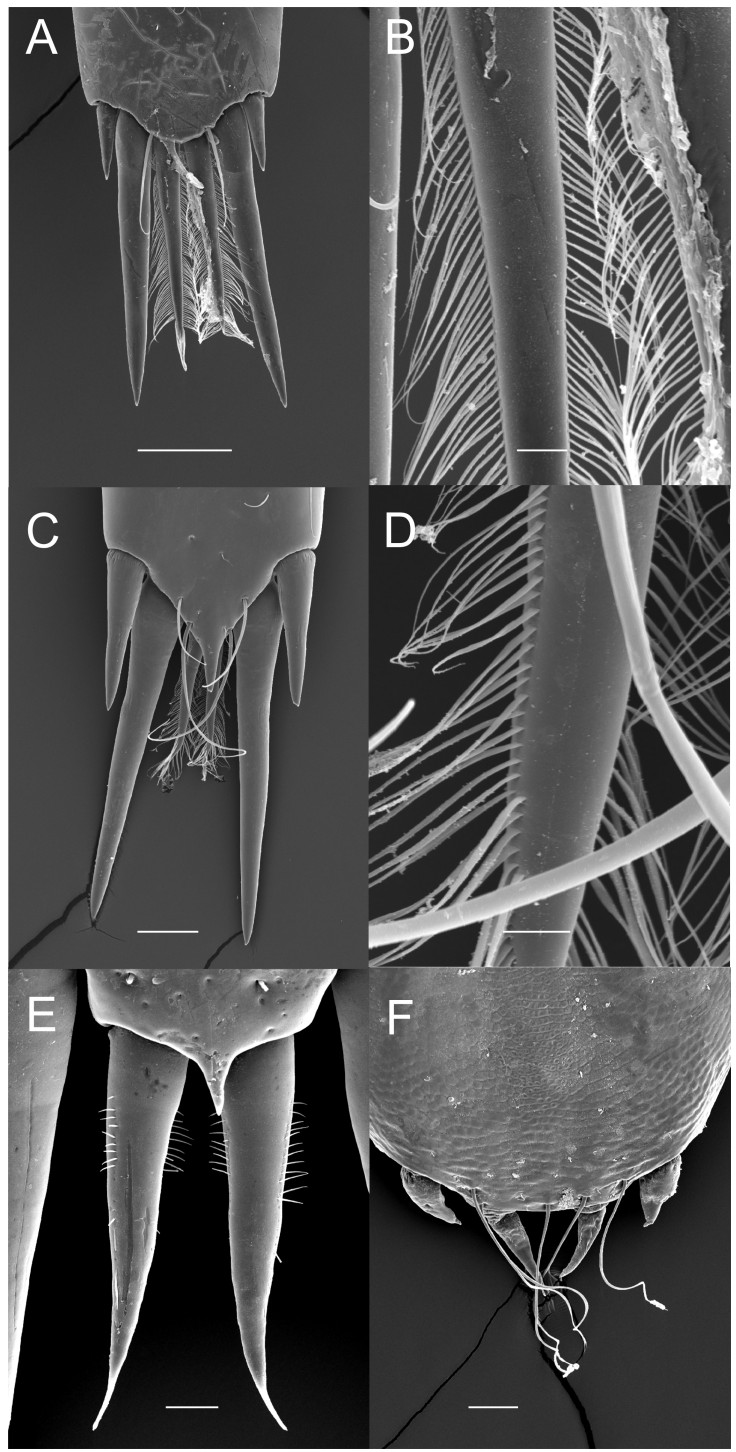

**Figure 4 Ornamentation of the posterior telson margin of some Pontoniinae.** (A) *Jocaste lucina*; (B) same, detail of median setae; (C) *Thaumastocaris streptopus*; (D) same, detail of median setae; (E) *Stegopontonia commensalis*; (F) *Conchodytes nipponensis*. Scale bars indicate 100 μm (A, C, F), 20 μm (E) or 10 μm (B, D).

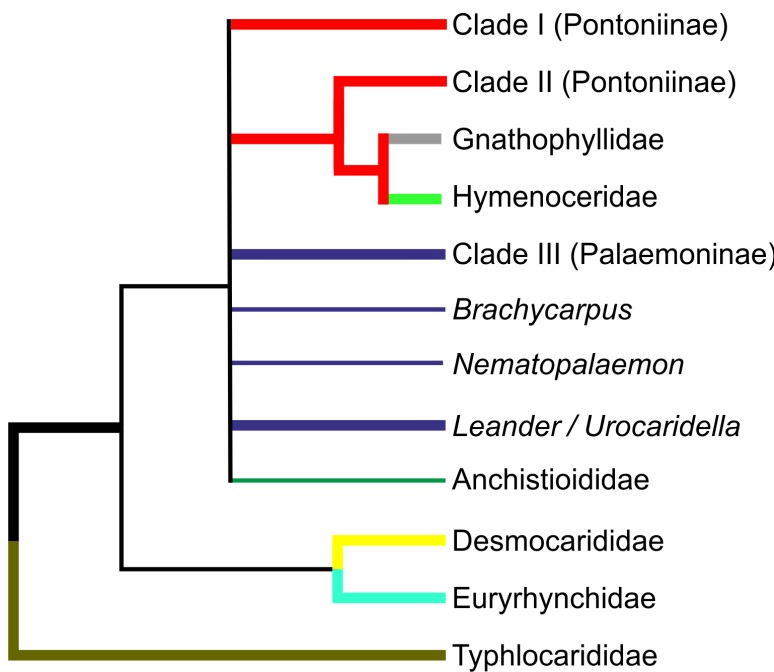

**Figure 5 Cladogram of hypothesised relationships of palaemonoid taxa derived from all molecular analyses.** Thicker lines denote where evidence is stronger.

either as sister or within the clade itself. There is very strong evidence that Gnathophyllidae and Hymenoceridae form a clade within Palaemonidae.

There is also strong molecular evidence that Palaemoninae and Pontoniinae do not form reciprocally monophyletic clades. Even when Gnathophyllidae and Hymenoceridae are considered as part of Pontoniinae, the evidence against this is strong, however the evidence against legitimate clades of Palaemoninae and Pontoniinae/Gnathophyllidae/Hymenoceridae is reduced markedly. Because of this, and because there are few instances when some species of Palaemoninae and Pontoniinae form strong clades with each other relative to others from their respective subfamilies, it remains possible that the addition of more markers and taxa could theoretically bring together reciprocal monophyletic clades that equate to Palaemoninae and Pontoniinae/Gnathophyllidae/Hymenoceridae. This does not however seem particularly likely.

Our molecular results mirror those of *Mitsuhashi et al. (2007)*, *Kou et al. (2013a)* and *Gan et al. (2015)*, who each recover a paraphyletic clade comprising the family Palaemonidae and Gnathophyllidae/Hymenoceridae. As already discussed by *Mitsuhashi et al. (2007)*, this relationship is underpinned by similarities in larval morphology. *Bruce (1986)* when describing the first zoea of *Gnathophyllum americanum* already remarked that they are fundamentally pontoniine in nature, and further highlighted the uniformity in larval form within Palaemonidae *sensu lato* when he described the zoea of *Hymenocera picta* (see *Bruce, 1988*). Recently, *Meyer et al. (2014)* also commented on the close morphological similarity between gnathophyllids, hymenocerids and pontoniines when describing the fine features under SEM of the zoea of *Gnathophyllum elegans*.

As regards the adults, *Holthuis (1955)*, *Holthuis (1993)* and *Chace (1992)* characterised both Gnathophyllidae and Hymenoceridae by the mandible with a vestigial or absent incisor process, third maxilliped being broadened, at least proximally (sometimes operculate) and the first maxilliped with the caridean lobe not distinctly overreaching the endite. The other palaemonoid families are therein jointly defined primarily by the slender third maxilliped and the mandible usually with a prominent incisor. Ample evidence exists that the vestigial or absent incisor is not a synapomorphy of these families. *Bruce (1986)* already commented that some species in the pontoniine genera *Periclimenaeus*, *Onycocaris* and *Typton* also lack an incisor, for example *Typton gnathophylloides* (see *Holthuis, 1951*, Plate 50). Conversely, some gnathophyllids, like *Pycnocaris chagoae*, harbour a rudimentary incisor (see *Bruce, 1972*, Fig. 5A). In fact, even in *Gnathophyllum elegans*, the type species of the family Gnathophyllidae, a vestigial incisor is present (see *Ashelby, De Grave & Johnson, 2015*, Fig. 5A). Although the third maxilliped is markedly operculate in *Gnathophyllum*, this is not the case for all gnathophyllid genera. The third maxilliped in *Gnathophylloides* is broadened, but not operculate (see *Bruce, 1973*, Fig. 4C), whilst only basally broadened in *Levicaris* (see *Bruce, 1973*, Fig. 8G). Conversely, some pontoniine genera equally have a much broadened third maxilliped, notably members of the genus *Conchodytes* (see *Fransen, 1994*, Figs. 35–37). The extensive bauplan modifications of the first maxilliped in pontoniine shrimps makes a comparison futile, perhaps the reason why this character distinction was not listed in the latest definition of the families by *Wicksten (2010)*. It should be noted that both *Hymenocera* and *Phyllognathia* do share a unique synapomorphy amongst palaemonoid shrimps, namely the basis of the third maxilliped being distinct from the ischiomerus, which in turn is marked by a distinct suture, delineating the ischium and merus. Both *Chace (1992)* and *Holthuis (1993)* used this character to separate the Hymenoceridae from the Gnathophyllidae. The current molecular analysis, as well as those of *Mitsuhashi et al. (2007)*, *Kou et al. (2013a)* and *Gan et al. (2015)* do demonstrate this not to be of familial importance.

In view of the overwhelming molecular evidence, the similarity in larval morphology and the weak morphological basis on which to separate adults into their respective families, Gnathophyllidae Dana, 1852 and Hymenoceridae Ortmann, 1890 are thus herein formally synonymised with Palaemonidae Rafinesque, 1815. As a result, the genera *Gnathophylleptum* d'Udekem d'Acoz, 2001, *Gnathophylloides* Schmitt, 1933, *Gnathophyllum* Latreille, 1819, *Levicaris Bruce, 1973*, and *Pycnocaris Bruce, 1972* (all formerly in Gnathophyllidae), as well as *Hymenocera* Latreille, 1819 and *Phyllognathia* Borradaile, 1915 (both formerly in Hymenoceridae), and their constituent species (see *De Grave & Fransen, 2011* for a listing) are now to be considered genera in Palaemonidae.

Our molecular analyses do not recover the two subfamilies within Palaemonidae, viz. Palaemoninae and Pontoniinae as reciprocally monophyletic clades. Instead, there appear to be at least two clades of Pontoniinae species (Clades I and II as per *Kou et al., 2013b*; *Gan et al., 2015*), including the ex-gnathophyllid and hymenocerid genera in Clade II, and yet these two clades of Pontoniinae do not necessarily form a clade with each other (Fig. 1). Within Palaemoninae, there is generally one large strongly supported clade of

species (here called palaemonid Clade III) (Fig. 1), which usually includes a couple more divergent species (*Palaemon concinnus, Palaemon elegans*). There are also a number of species of Palaemoninae which do not form a clade with other members of the subfamily, namely *Brachycarpus biunguiculatus, Nematopalaemon tenuipes*, and a clade of *Leander tenuicornis/Urocaridella pulchella*. When species of Palaemonidae are constrained to a clade without subfamily constraints, *Leander tenuicornis/Urocaridella pulchella* and *Nematopalaemon tenuipes* form a clade with the pontoniine Clade I (Fig. S2). These results do mirror the actual trees presented in *Kou et al. (2013a)* which equally do not show the two subfamilies to form monophyletic clades, although not discussed therein. Earlier, *Bracken, De Grave & Felder (2009)* had already hinted at the fact that the family as then defined was either para- or polyphyletic and the position of several pontoniine genera in their analysis was at odds with their current classification.

As already mentioned, the sole morphological character on which placement of a given genus in their respective subfamily is based is the ornamentation of the posterior margin of the telson, specifically the cuticular extensions. The terminology of these structures has been confusing in taxonomic descriptions, variously referred to as "spines", "stout setae", "spiniform setae". Herein, we adhere to the definition of *Watling (1989)* that a "spine" is a non-articulated, cuticular extension, with a "seta" being articulated, although we do acknowledge that non-articulated setae exist (see *Garm & Watling, 2013*), but these do not enter the discussion here. Following the classification of setal types by *Garm (2004)*, it is clear (Figs. 2–4) that the plesiomorphic condition in the family Palaemonidae comprises of a lateral pair of cuspidate setae, a submedian pair of elongated cuspidate setae and a median (or more) pair of plumose setae. Variations on this theme abound, with the median pair of plumose setae being thin and long to short(er) and stout, but nevertheless with a clear double row of poorly articulated setules on the shaft, thus still fitting the definition of plumose setae. In some taxa (Figs. 4E and 4F), the setules are reduced and the general appearance of the setae approaches that of cuspidate setae. Although cuspidate setae are known to occasionally have small outgrowths on their shaft, these are in the shape of denticles (*Garm, 2004*; *Garm & Watling, 2013*). We therefore interpret these median setae as reduced plumose setae.

Of course, concomitant with the rich bauplan diversity of pontoniine and palaemonine taxa, more variation exists than herein illustrated. For example, in *Hamopontonia*, the distal margin of the telson is emarginated and devoid of cuspidate and plumose setae; instead a number of simple setae are present (see *Bruce, 1970*) and in *Yeminicaris*, the distal margin is broadly rounded and devoid of cuspidate and plumose setae (see *Bruce, 1997*). A further example is illustrated in Fig. 10F, *Palaemon modestus*, where the median plumose setae are absent, the latter being characteristic for species of *Palaemon* previously considered to be a separate genus, *Exopalaemon* (see *De Grave & Ashelby, 2013*).

Nevertheless, from the evidence presented herein (Figs. 2–4) it is abundantly clear that the sole morphological character separating the two subfamilies does not hold true. In light of this, and supported by the molecular analyses, the subfamilies Palaemoninae Rafinesque

1815 and Pontoniinae *Kingsley, 1879* are herein formally synonymised and subfamilies are thus no longer recognised in Palaemonidae Rafinesque, 1815.

As in previous analyses (*Mitsuhashi et al., 2007*; *Bracken, De Grave & Felder, 2009*; *Kou et al., 2013a*; *Gan et al., 2015*) the position of Anchistiodidae remains uncertain, although it is clear that the family is closely related to Palaemonidae as herein defined. Historically the sole genus in this family, *Anchistioides* was often considered to be in Pontoniinae (now Palaemonidae), for instance by *Kemp (1922)*, *Gordon (1935)* and *Holthuis (1955)*. In more recent treatments, following *Chace (1992)* separate familial status has been the norm. *Chace & Bruce (1993)* remarked that the genus differs little from some pontoniines, separated only by seemingly minor adult morphological characters, but as pointed out by *Chace (1992)* and *Chace & Bruce (1993)*, the larvae, described by *Gurney (1936)* and *Gurney (1938)* differ sufficiently to support a separate family. As we cannot clarify the position of the genus *Anchistioides*, we refrain from analysing the morphological evidence and leave the family Anchistioididae as valid, until further evidence becomes available.

## ACKNOWLEDGEMENTS

Bregje W. Brinkmann and Cessa Rauch (Naturalis Biodiversity Centre) are acknowledged for sequencing the H3 genes of *Gnathophylloides mineri*, *Manipontonia psamathe* and *Pontonia manningi*.

### Funding

Part of the present work was supported by a research grant (project no. 41476146) from the National Natural Science Foundation of China (NSFC). The funders had no role in study design, data collection and analysis, decision to publish, or preparation of the manuscript.

### Grant Disclosures

The following grant information was disclosed by the authors:
National Natural Science Foundation of China (NSFC): 41476146.

### Competing Interests

The authors declare there are no competing interests.

### Author Contributions

- Sammy De Grave, Charles H.J.M. Fransen and Timothy J. Page conceived and designed the experiments, performed the experiments, analyzed the data, contributed reagents/materials/analysis tools, wrote the paper, prepared figures and/or tables, reviewed drafts of the paper.

### DNA Deposition

The following information was supplied regarding the deposition of DNA sequences:
GenBank (see Table 1 for accession numbers).

## Supplemental Information

Supplemental information for this article can be found online at http://dx.doi.org/10.7717/peerj.1167#supplemental-information.

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
