# Peer review of "Let’s be pals again: major systematic changes in Palaemonidae (Crustacea: Decapoda)"

_PeerJ, doi:10.7717/peerj.1167_

## Round 0.1 · original submission · Minor Revisions

All three reviewers pretty much agree that this is a good piece of work and requires only minor modifications. I wonder if you can come up with a snappier title that captures the importance of this paper for a more generalist readership?

·

Basic reporting

Well written, good support for synonymy.

Experimental design

No comments

Validity of the findings

Well supported

Additional comments

Please stay in present or past tense within a paragraph: use "defined" in line 100 and "characterized" in line 324. Double-check your use of commas: no comma after analyses (line 240), coral dwelling (line 283), pontoniines (line 322), based (line 326). But add comma after "chagoae" (line 333), "as already mentioned" (line 375). Insert semicolon (;) after setae, one 394.

Reviewer 2 ·

Basic reporting

This study builds on a growing body of evidence regarding the validity of several families in the Palaemonoidea and the two subfamilies within the Palaemonidae. Crucially, this is the first study to examine the morphological as well as molecular evidence and the authors bring this to a logical conclusion in synonymising several taxa. The evidence is considered and presented in a clear and concise manner and the proposed systematic re-arrangements seem justified.

The paper is highly informative and well written. There are relatively few typographic that I have indicated in the attached version of the MS. All relevant literature has been included and all tables and figures are relevant and justified. The SEM figures are clear and of high quality.

Experimental design

The background to the study is clearly defined in the introduction and the combined morphological and molecular approach adopted is necessary to fully answer the questions over the delineation of the taxa considered.

Validity of the findings

The authors acknowledge that systematic changes can often be contentious and be met with resistance but in this case the systematic decisions taken appear justified and the authors have been suitably cautious over the status of taxa where the evidence is still ambiguous.

Additional comments

A few minor corrections have been made in the attached annotated MS.

Annotated reviews are not available for download in order to protect the identity of reviewers who chose to remain anonymous.

Reviewer 3 ·

Basic reporting

No comments.

Experimental design

No comments.

Validity of the findings

No comments.

Additional comments

This is a very important article for caridean shrimp taxonomy. I agree with the result on the systematic position of Pontoniinae, Gnathopyllidae and Hymenoceridae.

I have request to add explanation about some morphological characters.

First. In current morphological taxonomy, the gill formula, especially the presence or absence of pleurobranch on the third maxilliped, is used as a distinguishing character between Palaemoniane and Pontoniinae. It is better that the evaluation of this character will be added in the discussion (near that of telson).

Second. The prominent cornea is distributed in Hymenoceridae, Gnathophyllidae (except for Phycocaris) and a few genera of Pontoniinae. I hope that the author's opinion for this character will be mentioned.

---

## Round 0.2 · accepted · Accept

Great new title and thank you for responding so promptly.